Pentoxifylline decreases post-operative intra-abdominal adhesion formation in an animal model

Yang Ya-Lin
Lee Meng-Tse Gabriel
Lee Chien-Chang
Su Pei-I
Chi Chien-Yu
Liu Cheng-Heng
Wu Meng-Che
Yen Zui-Shen
Chen Shyr-Chyr scchen@ntu.edu.tw
Department of Emergency Medicine, National Taiwan University Hospital and College of Medicine , Taipei , Taiwan
Walsh Stewart
Electronic publication date: 2018 Aug 24
Publication date: 2018
Volume: 6
Electronic Location ID: e5434
Received 2018 Mar 2; Accepted 2018 Jul 17
Copyright: ©2018 Yang et al.
Copyright year: 2018
Copyright holder: Yang et al.
License: This is an open access article distributed under the terms of the Creative Commons Attribution License, which permits unrestricted use, distribution, reproduction and adaptation in any medium and for any purpose provided that it is properly attributed. For attribution, the original author(s), title, publication source (PeerJ) and either DOI or URL of the article must be cited.
License URL: https://creativecommons.org/licenses/by/4.0/

Keywords: Angiogenesis, Tissue plasminogen activator level, Pentoxifylline, Inflammation, Fibrosis, Intra-abdominal adhesion formation, Collagen deposistin, Fibrinolysis

Funding: Taiwan National Science Council NSC 101-2314-B-002-055-MY3 Taiwan National Ministry of Science and Technology MOST 104-2314-B-002-028 MOST 106-2811-B-002-048 This work is supported by Taiwan National Science Council grant NSC 101-2314-B-002-055-MY3, and Taiwan National Ministry of Science and Technology grants MOST 104-2314-B-002-028 and MOST 106-2811-B-002-048. The funders had no role in study design, data collection and analysis, decision to publish, or preparation of the manuscript.

==============================
Background

Intra-abdominal adhesions develop after nearly every abdominal surgery, commonly causing female infertility, chronic pelvic pain, and small bowel obstruction. Pentoxifylline (PTX) is a methylxanthine compound with immunomodulatory and antifibrotic properties. The aim of this study was to investigate whether PTX can reduce post-operative intra-abdominal adhesion formation via collagen deposition, tissue plasminogen activator (tPA) level, inflammation, angiogenesis, and fibrosis.

Methods

Seventy male BALB/c mice were randomized into one of three groups: (1) sham group without peritoneal adhesion model; (2) peritoneal adhesion model (PA group); (3) peritoneal adhesion model with PTX (100 mg/kg/day i.p.) administration was started on preoperative day 2 and continued daily (PA + PTX group). On postoperative day 3 and day 7, adhesions were assessed using the Lauder scoring system. Parietal peritoneum was obtained for histological evaluation with hematoxylin and eosin (HE) and picrosirius red staining. Fibrinolysis was analyzed by tPA protein levels in the peritoneum by ELISA. Immunohistological analysis was also conducted using markers for angiogenesis (ki67+/CD31+), inflammation (F4/80+) and fibrosis (FSP-1+ and α-SMA+). All the comparisons were made by comparing the PA group with the PTX treated PA group, and p < 0.05 was considered statistically significant.

Results

Intra-abdominal adhesions were markedly reduced by PTX treatment. Compared with the PA group, PTX treatment had lower adhesion scores than the PA group on both day 3 and day 7 (p < 0.05). Histological evaluations found that PTX treatment reduced collagen deposition and adhesion thickening. ELISA analysis showed that PTX treatment significantly increased the level of tPA in the peritoneum. In addition, in the immunohistological analysis, PTX treatment was found to significantly decrease the number of ki67+/CD31+ cells at the site of adhesion. Finally, we also observed that in the PTX treated group, there was a reduction in the expression of F4/80+, FSP-1+, and α-SMA+ cells at the site of adhesion.

Conclusion

PTX may decrease intra-abdominal adhesion formation via increasing peritoneal fibrinolytic activity, suppressing angiogenesis, decreasing collagen synthesis, and reducing peritoneal fibrosis. Our findings suggest that PTX can be used to decrease post-operative intra-abdominal adhesion formation.

Introduction

Post-operative intra-abdominal adhesion after laparotomy is a source of considerable morbidity. It is estimated that more than 90% of patients develop primary intra-abdominal adhesion after laparotomy. Post-operative adhesions affect the quality of life in millions of people worldwide, causing many different types of complications, including chronic pelvic or abdominal pain, small bowel obstructions (SBO), and even infertility (Arung, Meurisse & Detry, 2011; Liakakos et al., 2001) SBO is the most common complication of adhesion and is observed in up to 70% of patients undergoing laparotomy (Ellis, 1997; Menzies & Ellis, 1990; Ten Broek et al., 2013). Although less commonly observed, up to 20% of female infertility has been associated with post-operative adhesions (Luijendijk et al., 1996).

The pathogenesis of post-operative intra-abdominal adhesion is a complex process that involves inflammation, collagen related clot formation, angiogenesis, fibrinolysis, and tissue repairs which include epithelial-mesenchymal transition (EMT)/endothelial-mesenchymal transition (EndMT) or mesothelial -mesenchymal transition (MMT) (DiZerega, 1997; Hellebrekers & Kooistra, 2011; Homdahl & Ivarsson, 1999). The key area in adhesion formation is the surface lining of the peritoneum. Injury of the peritoneum leads to activation of coagulation cascade and an inflammatory response consisting of hyperemia, fluid exudation, recruitment of floating mesothelial cells, and release of white blood cells and platelets into the peritoneal cavity (DiZerega, 1997; DiZeregal & Campeau, 2001). Normal fibrinolytic activity usually prevents fibrinous attachments for three to four days, and mesothelial repair occurs in five to six days after surgery (DiZeregal & Campeau, 2001). Therefore, previous studies focused on the cellular events three to six days after the peritoneal injury.

Pentoxifylline (PTX), a non-specific phosphodiesterase inhibitors, has been used to improve the walking ability in patients with intermittent claudication (Ernst, 1994; Hood, Moher & Barber, 1996; Rossner & Muller, 1987). Previous animal studies have also demonstrated that PTX can reduce post-operative adhesion, but the biological mechanisms that were responsible have not been fully clarified (Durmus et al., 2011; Hung et al., 2008; Jafari-Sabet et al., 2015; Tarhan et al., 2006). Five separate mechanisms on how PTX can alter the essential components in adhesion formation have been proposed: (1) reduction of inflammation (Durmus et al., 2011; Pollice et al., 2001); (2) reduction of collagen synthesis (Chen et al., 1999); (3) reduction of angiogenesis (Amirkhosravi et al., 1998) (4) increased fibrinolysis by up-regulation of tissue plasminogen activator (tPA) expression (Tarhan et al., 2006); (5) reduced fibrosis (Durmus et al., 2011; Wen et al., 2017). However, it is unclear if PTX can decrease post-operative intra-abdominal adhesion formation by simultaneously altering all of the five proposed mechanisms. Therefore, we aimed to investigate the effects of PTX on peritoneum collagen expression, peritoneal tPA expression, peritoneum angiogenesis, inflammation, and peritoneal fibrosis.

Methods

Animals

Male BALB/c mice weighing 25–30 g (Charles River Laboratories, BioLasco, Taiwan) were maintained in a temperature and light-controlled room (12-hour light/dark cycle) and allowed free access to water and food. The experimental protocols were approved by the Institutional Animal Care and Use Committee (IACUCs) of the National Taiwan University Hospital (approval ID: 20120285). All the protocols are in adherence to the guidelines established in the Guide for the Care and Use of Laboratory Animals of the National Health Research Institutes.

Experimental design

Seventy BALB/c mice were randomly divided into three different groups: (1) sham group without peritoneal adhesion model (sham group, n = 8); (2) peritoneal adhesion model (PA group, n = 31); (3) peritoneal adhesion model with PTX (Trental) (100 mg/kg/day i.p. once daily) administration was started on preoperative day 2 and continued daily (PA + PTX group, n = 31). We used a slightly modified standard adhesion model (Oncel et al., 2001). The details for peritoneal adhesion model had been reported before (Lee et al., 2016).

Briefly, mice were anesthetized using 2% isoflurane in oxygen. The abdomen was then shaved and disinfected with povidone iodine. A 4 cm median laparotomy was performed to gain access to the abdominal cavity. In the peritoneal adhesion model (PA and PA + PTX group), mice were pooled and randomly underwent surgery. The cecum was gently externalized and abraded with 20 vertically reciprocal movements of dry gauze. The right abdominal sidewall was rubbed more aggressively than the cecum until punctate bleeding was observed. The injury sites were cleaned with physiological salt solution and covered the gauze, making sure that there was no active bleeding. The cecum was then placed back into the abdominal cavity and the surgical wound was sutured. For the sham group, only open laparotomy and closure was conducted and there is no abrading of cecum and abdominal wall.

The PTX group received 100 mg/kg of PTX from the left abdominal cavity, whereas the other two groups (group sham and PA) received 0.125 ml of physiological saline solution. For preventing postoperative pain, buprenorphine (0.05 mg/kg s.c., twice daily) was administered during the two postoperative days. Mice were placed under a warming lamp and observed until they recovered fully from anesthesia. Mice were monitored daily for signs of wound infection and general health condition periodically until three or seven days after surgery.

Adhesion score

Mice were euthanized on postoperative day 3 and day 7. The abdominal cavity was opened via a U-shaped incision. The adhesion score was evaluated and was performed by an observer blinded treatment, using the Lauder scoring system (Lauder et al., 2011). The adhesions were graded in a blinded fashion using the classification system described (Table 1).

Table 1 Scoring system for intra-abdominal adhesion.

Score	Adhesion grading scale	
0	No adhesion	
1	Thin filmy adhesion	
2	More than one thin adhesion	
3	Thick adhesion with focal point	
4	Thick adhesion with planar attachment	
5	Very thick vascular adhesion or more than one planar adhesion	

Histology staining

Tissue samples from the parietal peritoneum, liver, and mesentery were collected after euthanasia. For histological staining, tissues were fixed in 10% neutral-buffered formalin (NBF), paraffin embedded, thinly sectioned. Tissue sections of 4–5 µm thickness were prepared for staining. After deparaffinization and rehydration, sections were counterstained in Gill’s hematoxylin (Sigma, St Louis, MO, USA) and for 5 min, cleared in 0.1% acid alcohol for 30 s, and rinsed in tap water, then stained in eosin (Sigma) for 2 min, cleared in 95% alcohol, and rinsed in 70% alcohol to remove the staining solution, dehydrated, and mounted for histologic assessment.

Picrosirius red staining

Picrosirius red staining was used to compare collagen and fibrosis in tissues between different groups. Peritoneal sections (4–5 µm) were deparaffinized, rehydrated and then subjection to counterstaining in Gill’s hematoxylin (Sigma) for 5 min, cleared in 0.1% acid alcohol for 30 s, and rinsed in running tap water. Then they were stained in Picrosirius Red Stain kit (Polysciences, Inc., Warrington, PA, USA). Subsequently, sections were dehydrated and mounted for assessment.

Tissue plasminogen activator

Peritoneal tissues were prepared by grinding on ice in radioimmunoprecipitation assay buffer (RIPA buffer) with protease inhibitor cocktail (Sigma). After the samples were centrifuged at 12,000 g for 15 min at 4 °C, supernatants were aspirated and placed in new tubes. Samples were analyzed for total antigen concentration of tPA, using commercially available ELISA kits from Molecular Innovations (Molecular Innovations, Novi, MI, USA). Total protein content was determined by Bradford assay (Sigma).

Immunohistochemistry

Formalin-fixed and paraffin-embedded peritoneal tissue was sectioned at 4–5 µm and then subjected to double immunostaining. Briefly, sections were deparaffinized, rehydrated and endogenous peroxidase activity was quenched by 3% hydrogen peroxide (H2O2) for 10 min. Sections were subjected to antigen retrieval performed in pH 6.0 citrate buffer using a microwave oven for 15 min. Blocking of non-specific binding was done by incubation with 2.5% horse serum at room temperature for 30 min. Sections were incubated with primary antibodies, rabbit anti-ki67 (1:200, Abcam, Cambridge, MA, USA), rat anti-F4/80 (1:200, Abcam) or rabbit anti-FSP-1 (1:200, Abcam) overnight at 4 °C. After washing with Tris-buffered saline (TBS, pH7.4), sections were incubated using the ImmPRESS AP anti-rabbit polymer reagent (Vector Laboratories, Burlingame, CA, USA) for 30 min at room temperature. Positive signals resulted in blue nuclear staining with the VECTOR Blue kit (Vector Laboratories). After washing and blocking again, the sections were incubated with goat anti-CD31 (1:500, R&D System, Minneapolis, MN, USA) for 1 h at room temperature. After washing, ImmPRESS HRP anti-goat polymer reagent (Vector Laboratories) was used for 30 min at room temperature. Positive reactions for endothelial cells resulted in brown red staining with the NovaRed substrate kit (Vector Laboratories). Sections were examined by light microscopy (Nikon Instruments, Nikon Corporation, Tokyo, Japan).

Immunofluorescence staining

Peritoneal tissue sections (4–5 µm) were performed for double immunofluorescence staining. Briefly, sections were deparaffinized, rehydrated and were treated with 0.3% H2O2 for 10 min to block endogenous peroxide activity and boiled in pH 6.0 citrate buffer using a microwave oven for 15 min. Sections were subsequently incubated with 5% Donkey serum for 20 min at room temperature. Sections were incubated with rabbit anti-cytokeratin 18 (CK18) (1:200, Enogene, New York, NY, USA) overnight at 4 °C, washed in phosphate-buffered saline (PBS, pH7.4) and incubated using donkey anti-rabbit DyLight 488 antibody (Thermo Scientific, Rockford, IL, USA) and Cy3-conjugated mouse anti-α-smooth muscle actin (α-SMA) (Sigma) and then mounted and subjected to fluorescence microscopy (Leica DMRA, Leica Microsystems, Wetzlar, Germany). Images were recorded at x100, x200 and x400 magnification of light microscopy, and were then digitalized and analyzed using Image-Pro Plus 6.0 software (Media Cybernetics, Rockville, MD, USA).

Statistical analysis

Normal distributed continuous data were expressed as mean ± standard error (SE). For non-parametric data, results were expressed as median ± interquartile range (IQR). The difference between continuous variables were evaluated using one-way ANOVA when data distribution was normal and a Mann–Whitney test was used for non-normal distributed continuous data. A p value of less than 0.05 was considered statistically significant. The statistical analyses were performed with GraphPad Prism (version 6.0, GraphPad Software, Inc., La Jolla, CA, USA).

Results

Deaths of animal

Surgical procedures were successfully completed on 69 animals, except for one mouse in the sham group, which died due to anesthesia-related complications before the commencement of surgery. One mouse in the PA + PTX group died during recovery from anesthesia. Four mice died within 48 h of surgery and were excluded from the study (PA and PA + PTX group, n = 2/group). Two mice in the PA group were excluded due to severe distress, according to three criteria in Health Evaluation of Experimental Laboratory Mice: very rough hair coat, hunched, and not eating or drinking. No mice in the PA + PTX treated group incurred any life-threatening side effects or deaths at 48 h after surgery, which would have lead to exclusion from the study. Therefore, a total of eight animals were excluded from the study.

Mice were euthanatized for the planned experiments on postoperative day 3 (PA group, n = 14; PA + PTX group, n = 13) and day 7 (sham group, n = 7; PA group, n = 13; PA + PTX group, n = 15).

Pentoxifylline treatment reduces adhesion score

Total adhesion scores data was examined and plotted for post-operative day 3 and 7 (Fig. 1). We observed that the sham group, which had not undergone the adhesion model, had a significantly lower adhesion score than the animals that had undergone adhesion model, as expected. The PA + PTX group (median, 1.00; IQR, 0.50–2.00) had a significantly lower adhesion score than the PA group (median, 2.00; IQR, 2.00–3.00) on day 3 (p < 0.05). On day 7, mice treated with PTX (median, 3.00; IQR, 0.00–3.00) still had lower adhesion scores than the PA group (median, 3.00; IQR, 3.00–4.50) (p < 0.05).

Figure 1 Intra-abdominal adhesion score.

The PA+PTX group had a lower adhesion score. Data are expressed as the median ± IQR. ∗P < 0.05, ∗∗P < 0.01, ∗∗∗P < 0.001, respectively.

Pentoxifylline treatment inhibits collagen deposition

We used the HE staining to compare changes in peritoneal structure (Fig. 2A). In general, there was increased thickness of the submesothelial layer on day 3, and the adhesion score also increased. We observed that the sham group had the thin submesothelial layer as demonstrated in Fig. 2A. The severe adhesion and thick submesothelial layer were observed, as well as the increased cellularity in the PA group. In contrast, the PA + PTX group had less peritoneal submesothelial thickness and adhesion severity as compared with the PA group.

Figure 2 Pentoxifylline treatment inhibits collagen deposition.

(A) Representative images of HE staining. No adhesion was observed in the sham group. Severe liver or bowel adhesion were observed in the PA group, whereas the PTX treated group had decreased adhesion severity. (B) Representative images of Picrosirius red staining. The thickness of the collagen deposition was increased in the PA group, whereas the PTX treated group has less collagen deposition. (Original magnification, x200, bar = 100 µm).

We further used Picrosirius red staining to assess the quality of collagen fiber in peritoneal adhesion (Fig. 2B). Compared with the PA group, the PA + PTX group had less collagen deposition and thickness of the abdominal adhesions. Our data suggested that PTX could decrease collagen deposition during adhesion formation.

Pentoxifylline treatment increased tPA level

The tPA protein levels in the peritoneum of mice were perceived to be measured and plotted on post-operative day 3 and 7 (Fig. 3). We observed that the sham group of mice had the lowest tPA protein level throughout the study period (p < 0.001). Those mice treated with PTX had higher tPA protein levels than untreated mice (the sham and PA groups). There was significant difference between the PA + PTX group and the PA group on day 7 (0.365 ± 0.024 vs. 0.193 ± 0.03, p < 0.001).

Figure 3 The levels of tPA protein in peritoneum.

tPA protein level increased in both PA group and PA+PTX group on postoperative day 3 and day 7. PTX-treated mice increased tPA protein level at postoperative day 7 compared with postoperative day 3. Data are expressed as the mean ± SE. ∗P < 0.05, ∗∗P < 0.01, ∗∗∗P < 0.001, respectively.

Figure 4 Pentoxifylline treatment reduced angiogenesis.

(A) Representative examples of double immunohistochemistry staining of ki67 (blue) and CD31 (brown-red) (arrows) in peritoneum on day 3 and day 7. (Original magnification, x200, bar = 100 µm). (B) The graph shows numbers of cells expressing Ki67+ and CD31+ (proliferating endothelial cells) on day 3 and day 7. (C) The graph shows the percentage of CD31+ vessel area per field at x200 magnification on day 3 and day 7. Data are expressed as the mean ± SE. ∗P < 0.05, ∗∗P < 0.01, ∗∗∗P < 0.001, respectively.

Pentoxifylline treatment reduced angiogenesis

We performed immunohistochemical staining assay to analyze the status of angiogenesis during peritoneal repair. Proliferating endothelial cells were identified as those cells with cytoplasmic CD31 staining and nuclear Ki67 staining (Fig. 4A, arrows). Cells that stained positive for CD31 but without nuclear staining for Ki67 were scored as nonproliferating endothelial cells. We examined the effect of endothelial cell proliferation in peritoneum was quantified by measuring the number of ki67+ CD31+ cell at the site of adhesion (Fig. 4B). Our results showed that the PA group had significantly higher proliferating endothelial cells compared to both the sham and PA + PTX groups on day 3 and day 7 (all p < 0.001). We also observed that the ki67+ CD31+ proliferating endothelial cell count decreased substantially over time (p < 0.001) in the PA group. Also, we examined the angiogenic effect in peritoneum by measuring the area of CD31+ microvessel at the site of adhesion (Fig. 4C). We also observed that the PA group had a vessel coverage (the percentage of area covered by CD31+ per field) of 1.63% at day 3, and 3.63% at day 7, respectively. On post-operative day 7, the PA group demonstrated an increase in CD31 expression compared to day 3 and indicated that blood vessel formation was significantly more prominent in the PA group compared to the PA + PTX groups (p < 0.001). Thus, PTX can significantly suppress angiogenesis during peritoneal repair.

Pentoxifylline treatment reduced inflammation

Inflammation, an important component both in normal and pathological healing, is a protective response to tissue injury, designed for removal of the causative agent and restoration of tissue structure and function. We performed immunohistochemical staining assay to analyze the infiltration of macrophage during peritoneal repair. F4/80 is a macrophage-specific marker in mice. As shown in Fig. 5A, a large number of F4/80+-expressed cells were observed in the PA group. Quantification analysis of the IHC image revealed significantly increased expression of F4/80+ cells on day 3 (0.80% ± 0.10%), and highest expression on day 7 (2.56% ± 0.22%) in the PA group as compared to the sham group (p < 0.01 for both day 3 and day 7) or the PA + PTX group (1.37% ± 0.28%, p < 0.01 on day 7, Fig. 5B). The PA + PTX group had significantly lower expression of F4/80+ as compared with the PA group.

Figure 5 Pentoxifylline treatment reduced inflammation.

(A) Immunohistochemistry for F4/80 was performed on mice peritoneal tissue in the different groups at day 3 and day 7. The F4/80 expression was increased in the PA group. Representative images of the sham group, the PA group, and the PA+PTX group are shown (Original magnification, x200, bar = 100 µm). (B) Quantification of F4/80+ cells (%) in high-powered field (HPF) at x400 magnification. Data are expressed as the mean± SE. ∗P < 0.05, ∗∗P < 0.01, ∗∗∗P < 0.001, respectively.

Pentoxifylline treatment reduced the expression of fibrosis marker FSP-1

FSP-1, also known as fibroblast-specific protein 1 (FSP1), belongs to the S100 superfamily of cytoplasmic calcium-binding proteins and can be expressed by cell of mesenchymal origin or fibroblastic phenotype. This protein is reported to be specific for fibroblasts and plays a causal role in EMT. As shown in Figs. 6A and 6B, a large number of FSP-1+-expressed cells were observed in the PA group. Quantification analysis of the IHC image revealed significantly increased expression of FSP-1+ on day 3 (5.37% ± 1.03%), and highest expression on day 7 (11.26% ± 1.66%) in the PA group compared to the PA + PTX group (p < 0.05 for both day 3 and day 7, Fig. 6C). Consistently, we found the mice treated with PTX had significantly reduced expression of FSP+ as compared with the PA group.

Figure 6 Pentoxifylline treatment reduced the expression of fibrosis marker FSP1.

(A–B) Immunohistochemistry for FSP-1 was performed on mice peritoneal tissue in the different groups at day 3 and day 7. The FSP-1 expression was increased in the PA group. Representative images of the sham group, the PA group, and the PA+PTX group are shown. (Original magnification, x100, bar = 100 µm). (C) Quantification of FSP-1+ cells (%) in high-powered field (HPF) at x400 magnification. Data are expressed as the mean±SE. ∗P < 0.05, ∗∗P < 0.01, ∗∗∗P < 0.001, respectively.

Figure 7 Pentoxifylline treatment reduced the expression of fibrosis marker α-SMA.

(A–B) Double immunofluorescence was performed with CK18 and α-SMA in mice peritoneum in the different groups at day 3 and day 7. Immunofluorescence shows the staining of mesothelial cells by CK18 was expressed in green color, and myofibroblast by α-SMA was expressed in red color. In the PA group, we observed a few CK18+ cells were co-localized with α-SMA in the mesothelial layer. (Original magnification, x400, bar = 100 µm). (C) Quantification of α-SMA+ cells (%) in high-powered field (HPF) at x400 magnification. Data are expressed as the mean± SE. ∗P < 0.05, ∗∗P < 0.01, ∗∗∗P < 0.001, respectively.

Pentoxifylline treatment reduced the expression of fibrosis marker α-SMA

As a response to injury, mesothelium undergoes a change called mesothelial-to-mesenchymal transition (MMT). Thus, we further performed double immunofluorescence staining for CK18 and α-SMA for peritoneal injury (Figs. 6A and 6B). Many studies have demonstrated that mesenchymal cell markers, including α-SMA, are proposed as indicators of EMT (Margetts et al., 2005). Cytokeratin (CK) are structural marker proteins specific for epithelial cells, and CK18 is highly expressed in mesothelial cells. α-SMA has become the most reliable marker of myofibroblasts. Figure 7A is consistent with previous studies, where the PA group observed a few CK18+α-SMA+ double-positive cells appear first in the mesothelial monolayer and later in the reorganized submesothelial matrix. We examined the extent of accumulation of myofibroblasts in peritoneum was quantified by assessing the percentage α-SMA+ cells (Fig. 7C). Our result showed that the PA group had significantly increased α-SMA+ expression at day 3 (3.48% ± 1.28%), and highest expression at day 7 (13.71% ± 1.40%) compared with the PA + PTX group (p < 0.05 for day 3 and p < 0.01 for day 7, respectively). PTX significantly attenuated thickening of fibrotic peritoneum, and accumulation of α SMA+ myofibroblasts in peritoneum after injury.

Discussion

In this study, we demonstrated that PTX treatment could effectively reduce post-operative intra-abdominal adhesion formation. PTX could prevent peritoneum adhesion formation via five related biological processes: increasing fibrinolysis, reducing inflammation, reducing angiogenesis, reducing collagen deposition, and reducing fibrosis.

Post-operative intra-abdominal adhesion formation is considered to be an inevitable result of peritoneum injury after abdominal surgery. Peritoneum injury initiates an inflammatory response, which increases vascular permeability leading to fibrin release and adhesion formation (DiZeregal & Campeau, 2001). Under normal conditions, the majority of fibrin is degraded within a few days by locally released proteases of the fibrinolytic system (Harris, Morgan & Rodeheaver, 1995; Sulaiman et al., 2002). In a pathological state, if fibrinolysis does not occur within five to seven days of the peritoneal injury, the provisional fibrin matrix persists and more gradually becomes organized as the collagen-secreting myofibroblasts and other repairing cells infiltrate the matrix (Homdahl & Ivarsson, 1999). This process leads to peritoneal adhesion and new blood vessel formation (angiogenesis) (Saltzman et al., 1996).

We hypothesized that there are at least four mechanisms that PTX treatment might result in to reduce post-operative adhesion. First, PTX has been reported to alter rheological properties of blood such as: decreasing blood viscosity by stimulating fibrinolysis to reduce plasma fibrinogen concentrations, increasing erythrocyte flexibility and platelet deaggregation, and inhibiting neutrophil activity to reduce the tissue damage (McCarty, O’Keefe & DiNicolantonio, 2016). The alteration in the rheological properties of blood may be the reason why we observed that the tPA level was significantly higher in the PTX treated group than those without PTX treatment. In fact, we have previously found that mice treated with therapeutic hypothermia have increased tPA levels and reduced post-operative adhesion (Lee et al., 2016).

Second, the anti-inflammation property of PTX has been well established by several previous studies, and has been found to attenuate the cardiopulmonary bypass (CPB)-induced systemic inflammatory response syndrome and postoperative mortality (Barkhordari et al., 2011; Heinze et al., 2007; Otani et al., 2008). PTX has been found to affect inflammation by reducing the plasma levels of pro-inflammatory cytokines such as TNF-α, IL-1 and IL-6 (Otani et al., 2008; Pollice et al., 2001). The reduction of cytokines at the site of injury may explain why we observed a reduction in the infiltration of macrophages in the PTX treated group.

Third, PTX also has been reported by previous studies to inhibit endothelial cell proliferation and angiogenesis (Gude et al., 2001; Hasebe, Thomson & Dorey, 2000). Vlahos et al. (2010) reported that PTX might cause suppression of endometriotic lesions by suppressing angiogenesis through VEGF-C and flk-1 expression. Recent evidence also found that PTX inhibits PKC-dependent activation of NF κB and prevent hypoxia-induced expression of VEGF (Amirkhosravi et al., 1998). Our results on reduction in angiogenesis in PTX treated groups correspond with the above findings.

Fourth, PTX was reported by previous studies to down regulate the intracellular signaling of TGF-β; which can affect collagen synthesis and fibrosis through the cAMP–PKA pathway (Fang et al., 2000; Kucich et al., 2000). Through PKA, PTX has been found to reduce TGF-β-induced collagen synthesis in vascular smooth muscle cells and human peritoneal mesothelial cells (Chen et al., 1999; Hung et al., 2003). This might explain why we observed a lower amount of collagen deposition in PTX treated mice. Moreover, TGF-β1 has been reported to be the key initiating factor of fibrosis, and is also known to strongly induce EMT or EndMT (Lamouille, Xu & Derynck, 2014; Piera-Velazquez, Li & Jimenez, 2011). EMT is defined as a cellular and molecular change that is usually characterized by loss of cell–cell adhesion, the down-regulation of E-cadherin and other epithelial genes, accompanied by the acquisition of mesenchymal cell morphology, increased contractility and actin stress fibers. This might explain why we observed reduction in markers of fibrosis with PTX treatment.

In this study, we found that PTX treatment decreased intra-abdominal adhesion formation by reducing fibrosis, but it was not in our initial objective to confirm whether the reduction in fibrosis might affect general wound healing. The main reason is because several studies have already found that PTX can instead improve general wound healing. Parra-Membrives et al. (2007) showed that PTX improved healing of experimental ischemic colorectal anastomoses by reducing wound and intra-abdominal infections, adhesion formation, and leaks. Comert et al. (2000) showed PTX has a positive effect on the obstructive jaundice caused by intestinal anastomosis healing by suppressing endotoxin-induced TNF-α release from macrophages and monocytes, and by having a stabilizing effect on the neutrophils. Therefore, future studies may need to clarify the time-frame on how PTX treatment can reduce fibrosis and yet improve wound healing, before a clinical trial of PTX can be recommended on post-operative patients. In addition, future studies should also clarify the mechanism on how streptokinase interact synergistically with PTX to reduce post-operative adhesion (Jafari-Sabet et al., 2015).

Conclusion

In conclusion, our study showed that PTX may decrease intra-abdominal adhesion formation via increasing peritoneal fibrinolytic activity, reducing inflammation, suppressing angiogenesis, decreasing collagen synthesis, fibroblast producing and peritoneal fibrosis. We believe that future studies should take into the account that PTX can reduce intra-abdominal adhesion formation through multiple pathways.

Supplemental Information

Supplemental Information 1 Original full-size images

Click here for additional data file.

Supplemental Information 2 Supplemental Tables

Results from statistical analyses.

Click here for additional data file.

Data S1 Raw data

Click here for additional data file.

We thank the staff of the Core Labs, the Department of Medical Research, and National Taiwan University Hospital for technical support.

Additional Information and Declarations

Competing Interests

Author Contributions

Animal Ethics

Data Availability

The authors declare there are no competing interests.

Ya-Lin Yang performed the experiments, analyzed the data, contributed reagents/materials/analysis tools, prepared figures and/or tables, authored or reviewed drafts of the paper, approved the final draft.

Meng-Tse Gabriel Lee analyzed the data, contributed reagents/materials/analysis tools, prepared figures and/or tables, authored or reviewed drafts of the paper, approved the final draft.

Chien-Chang Lee, Pei-I Su, Chien-Yu Chi, Cheng-Heng Liu, Meng-Che Wu and Zui-Shen Yen authored or reviewed drafts of the paper, approved the final draft.

Shyr-Chyr Chen conceived and designed the experiments, analyzed the data, authored or reviewed drafts of the paper, approved the final draft.

The following information was supplied relating to ethical approvals (i.e., approving body and any reference numbers):

The experimental protocols were approved by the Institutional Animal Care and Use Committee (IACUCs) of the National Taiwan University Hospital (approval ID: 20120285).

The following information was supplied regarding data availability:

The raw data are provided in the Supplemental Files.

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
