# Peer review of "Pentoxifylline decreases post-operative intra-abdominal adhesion formation in an animal model"

_PeerJ, doi:10.7717/peerj.5434_

## Round 0.1 · original submission · Minor Revisions

The reviewers have made a number of suggestions which would improve the manuscript. The manuscript should report the precise number of mice evaluated in each arm of the study. As the reviewers point out, 70 does not divide equally into three groups. Presumably there were some intraoperative or early deaths but the details should be provided.

·

Basic reporting

The study conduct and reporting was excellent, fulfilling all the required criteria.

Experimental design

The experimental design was excellent, fulfilling all the required criteria.

Validity of the findings

The main issue with this study, is the lack of correlation with general woudn healing. While adhesion reduction was demonstrated with PTX, could the same impact on fibrosis affect healing of abdominal wounds and anastomoses? Before clinical trial can be recommended this question needs to be carefully answered in animal models. This warrants specific discussion in the manuscript.

Additional comments

This is a great study, but some caution regarding wound healing effects needs to be exhibited and discussed, prior to simply recommending clinical trials.

Reviewer 2 ·

Basic reporting

Literature searching of about of prevention of adhesion with PTX should be done again. Some studies related to the subject are given below. These references should be included in the study.
1. Durmus AS, Yıldız H, Yaman M and Simsek H. 2011. The effects of heparin and pentoxifylline on prevention of intra-abdominal adhesions in rat uterine horn models: histopathological and biochemical evaluations. Revue Med Vet 162(4): 198-203.
2. Jafari-Sabet M, Shishegar A, Saeedi AR and Ghahari S. 2015. Pentoxifylline increases antiadhesion effect of streptokinase on postoperative adhesion formation: involvement of fibrinolytic pathway. Indian J Surg 77(3): S837–S842.

Experimental design

Line 32 and 104: Seventy male BALB/c mice were randomized into one of three groups: How were 70 mice divided into 3 equal groups? Number of "n" must be specified in the text.

Validity of the findings

no comment

Additional comments

Some spelling mistakes are given below.
1. Line 38: tissue tPA: “tissue” should be removed.
2. Line 276: Peritoneum Injury: Peritoneum injury.
3. Line 298: such as (TNF-α, IL-1 and IL-6): such as TNF-α, IL-1 and IL-6.

References should be re-controlled (References, journal names etc).
Some references are same.
Line 345: diZerega GS. and Line 347: diZerega GS.
Line 348: diZerega GS, and Campeau JD. and Line 350: diZeregal GS, and Campeau JD.

·

Basic reporting

Manuscript is well written and easily understandable. Literature is well referenced and relevant to the study. Regarding supplemental data, if the data is to be published, I would recommend it to be in a more user friendly format so that I could be understood by most readers even if they are not experts in interpreting statistical data. I would suggest data to be summarized in a table format.

Experimental design

Experimental design is well defined and seems to address the goal of the study appropriately. Scoring system and investigations to assess adhesion formation are detailed and extensive. However, I would suggest further clarification or revision in following areas of methodology:
a) In the experimental design (lines 103-118), it is not clear what was done to the Sham group of Mice. My interpretation is that the Sham group is the one without peritoneal adhesion. It is not clear whether this group underwent any kind of surgical intervention or not. Whether there was no intervention at all or this group had open laparotomy and closure without abrading the cecum, needs to be clarified.
b) Line 120 mentions that mice were euthanized on post operative days 3 and 7. I would suggest more detailed explanation on how many mice were euthanized on days 3 and 7 respectively, from each group. I assume out of the 3 groups created out of 70 mice, some would be euthanized on day 3 and others on day 7. I would suggest explanation of those numbers and whether they were randomized or not.

Validity of the findings

Study seems to be well structured and replicable. I would suggest a comment whether any of the experimental mice incurred any side effects (e.g death leading to exclusion from study) from pentoxyfilline administration.

Additional comments

Since Lauder scoring system was used to classify the severity of adhesions, a table detailing the Lauder system would be very beneficial for readers

---

## Round 0.2 · accepted · Accept

Thank you for addressing all the comments and submitting your revised manuscript. I confirm that it is now Acceptable

While you are in production, please can I ask you to make two very minor language corrections?

Results, Deaths of Animals. section

Lines 187 and 189 please correct the phrase 'one mice' to 'one mouse'